# Parental and peer support and modelling in relation to domain-specific physical activity participation in boys and girls from Germany

Anne K. Reimers[1]*, Steffen C. E. Schmidt[2], Yolanda Demetriou[3], Isabel Marzi[1], Alexander Woll[2]

1 Department of Sport Science and Sport, Friedrich-Alexander-University Erlangen-Nuremberg, Erlangen, Germany, 2 Institute of Sport and Sport Science, Karlsruhe Institute of Technology, Karlsruhe, Germany, 3 Department of Sport and Health Sciences, Technical University of Munich, Munich, Germany

* anne.reimers@fau.de

**Data Availability Statement:** Data cannot be shared publicly because of strict ethical conditions with which study investigators are obliged to comply: The Charité/Universitätsmedizin Berlin

## Abstract

### Background

Physical activity (PA) as a precondition of child development is related with social environmental correlates. However, domain-specific PA and gender issues have been neglected in studies on social support and modelling and PA in school-aged children. The aim of this study was to assess the relationships of parental and peer modelling and social support with domain-specific PA participation in a large sample of school-aged children, taking gender into account.

### Methods

3,505 school children aged 6 to 17 years old participated in the German nationwide 'MoMo' cohort-study. By using the MoMo-PAQ the participants and their parents provided self-report data on perceived social support and social modelling and domain-specific PA participation. Relationships of social environmental variables and the physical outcomes were analysed by logistic regression analyses.

### Results

At secondary school level, girls were less likely than boys to participate in physical activity in and outside of sports clubs, extra-curricular physical activity and in outdoor play ($p < 0.05$), but at primary school level this pattern only applied to club sport ($p < 0.01$). Girls also received less social support than boys ($p < 0.01$). Physical activity participation in all domains was associated with any of the social support and modelling variables and differences between physical activity domains and between boys and girls occurred. Most consistently physical activity in sports clubs was related with the social environmental correlates in boys (primary school: $R^2 = 0.60$; secondary school: $R^2 = 0.45$) and girls (primary school: $R^2 = 0.53$; secondary school: $R^2 = 0.47$).

ethics committee and the Federal Office for the Protection of Data explicitly forbid making the data publicly available because informed consent from study participants did not cover public deposition of data. However, the minimal data set underlying the findings is archived at the Institute of Sports and Sports Science of the Karlsruhe Institute of Technology (KIT) and can be accessed by interested researchers on site. On-site access should be submitted to the Institute of Sports and Sports Science, Karlsruhe Institute of Technology, Engler-Bunte-Ring 15, 76131 Karlsruhe, Germany (info@sport.kit.edu).

**Funding:** This present study is based on data from the "Motorik-Modul Longitudinal Study (MoMo) (2009 – 2021): Physical fitness and physical activity as determinants of health development in children and adolescents". MoMo is funded by the Federal Ministry of Education and Research (funding reference number: 01ER1503) within the research program ´long-term studies´ in public health research. We acknowledge support by Deutsche Forschungsgemeinschaft and Friedrich-Alexander-Universität Erlangen-Nürnberg (FAU) within the funding programme Open Access Publishing.

**Competing interests:** The authors have declared that no competing interests exist.

## Conclusions

In future, reciprocal relationships of social environmental variables and PA should be considered in longitudinal studies to obtain insights into the direction of the associations. Furthermore, interventions encompassing the social environment and focussed particularly on the promotion of domain-specific PA in girls in secondary school-age are warranted.

## Introduction

Physical activity and sports are essential protective factors for many non-communicable diseases like diabetes mellitus, and dementia and strongly related to cardio-metabolic biomarkers, physical fitness, bone health, quality of life, and psychological distress [1, 2]. Furthermore, regular physical activity can increase life-expectancy [3] and physical activity is a precondition for the motoric, cognitive, emotional and social development of children [4]. However, in Germany, only 17.4% of boys and 13.1% of girls comply with the physical activity guidelines of the World Health Organization [5], which recommend a minimum of 60 minutes of at least moderate physical activity per day [6].

There is a gap in levels of physical activity and physical activity participation between male and female children starting early in life and continuing through adulthood until old age [7–10]. In Germany, gender differences in physical activity participation favouring males can be observed already from the age of four onwards [5] and are also pronounced in later stages of life [11]. Gender-related social constructive theories such as "doing gender" or the socialisation theory [12] partly ascribe gender differences between boys and girls to social and cultural norms and socialisation processes. Although traditional and conventional ways of seeing boys and girls and even stereotypical gender roles are more flexible in the field of physical activity and sports, there are still predominant role concepts [13, 14]. Many sports and leisure-time physical activities do not comply with traditional images of femininity [15]. Thus, girls are less likely to engage in physical activity and sports activities–especially in seemingly typical masculine activities like playing soccer [16]–as this is not compatible with their learned female behavioural role [15].

Additionally, the social environment probably reacts differently to girls and boys: first, as many physical activities and sports are viewed as traditionally masculine (e.g., boxing or soccer) or feminine (e.g., synchronised swimming), parents, peers or caregivers are more likely to nudge children to choose activities that comply with their gender roles based on these stereotypes [17]. Second, the encouragement of significant reference persons to engage in sports and physical activity participation might be greater in boys than in girls, because physical activities in principle are rather assigned to masculine behaviour [18, 19]. Third, based on Social Learning Theory [20], children's behaviour and behavioural choices are also affected by social models, and it has been indicated that children are more likely to imitate behaviours of same-sex models [21]. The same-sex imitation hypothesis assumes that imitation of social models occurs as a function of credibility and relevance of the social model, so that children tendentially prefer imitation of the same-sex parent [21]. Thus, girls possibly have fewer same-sex models for physical activity because women and especially their mothers are less active than the fathers as available male role models for boys [22, 23].

Overall, parental and peer modelling and social support are considered to enable or to foster physical activity participation in children and youth. Parental modelling of activity is positively related to children's participation in physical activity like outdoor play, sports or walking

for transport [24–27]. For example, the provision of instrumental support like driving a child to places where they can play sports or buy equipment are necessary assistances to engage in some sporting activities. Concerning support from parents, evidence on positive relationships of tangible and intangible social support on youth's physical activity were obtained from many studies as summarised in review articles [28, 29]. In a study of fifth grade students from Germany, Schoeppe et al. [30] confirmed the same-sex imitation hypotheses by finding relationships between girls' leisure-time physical activity and maternal sport participation and between boys' activity and paternal sport participation. Similar results were found by Lijuan et al. [31] in Chinese children, by Kirby et al. [32] in Scottish adolescents, and by Cheng et al. [33] in Brazilian adolescents. Another study from Germany reported positive relationships of social support and social modelling from parents and moderate-to-vigorous physical activity (MVPA) in elementary school children and also emphasised gender differences in MVPA, but did not analyse parental influences on MVPA separately for both genders [27]. Schmiade and Mutz [34] also ascertained that social support and parental modelling predicted children's participation in organised sports courses in preschool children from Germany, but did not focus on differences between boys and girls. Focussing on gender differences in the provision of social support, Hoefer et al. [35] found parents being more likely to transport boys to physical activity locations than girls. However, in a recent study of 11-year old children from Shanghai [36] no gender differences in terms of logistic support and explicit modelling for physical activity of parents were observed.

Some studies also indicated the role of social support and social modelling from peers predicting physical activity in children and adolescents. The presence of peers and the presence of peers being physically active were associated with an increase in physical activity in a wide range of children and adolescents from the age of 3–5 to 15 years old [37–40]. In a study of adolescents from central England boys also perceived more peer support than girls [41].

In relation to the course from childhood to adolescence, it has been assumed that the significance of parental modelling and support declines in favour of an increase of the significance of peer modelling and support [42]. However, this has only been confirmed by few studies that simultaneously examined parental and peer influences on physical activity in relation to age groups [32, 38, 43].

In summary, until now a large body of scientific literature on the relationships of social support and social modelling with physical activity in children and adolescents is available. However, previous research has sparsely focussed on gender differences concerning this relationship. Little is known about gender differences in the provision of social support from parents and peers. Furthermore, previous studies analysing the effects of social support have mainly concentrated on overall or leisure time physical activity [29, 43], or on active transportation [44], but did not take a large variety of different domains of physical activity like physical activity in sports clubs, extra-curricular physical activity, and outdoor play into account [45]. Nevertheless, in a study on social influences on adolescent health-related physical activity in structured and unstructured settings Spink and colleagues [46] showed that in structured settings peer compliance predicts membership in high active groups, and in unstructured settings peer conformity is additionally relevant for participation in high active groups. Thus, we hypothesize that for specific domains different types and providers of social support and social modelling are relevant.

Taking the previous mentioned research gaps into account, the aim of this study is to examine gender differences in parental and peer support and modelling for physical activity participation, and the relationship of parent and peer social support and modelling with domain-specific physical activity participation in a large sample of children and adolescents from Germany.

## Material and methods

### Study design

The MoMo Study is a nationwide study on physical fitness and physical activity in German children and adolescents, and part of the German Health Interview and Examination Survey for Children and Adolescents, KiGGS [47, 48]. To ensure a diverse sample of German children and adolescents, a nationwide, stratified, multi-stage sample with three evaluation levels was drawn [49]. First, a systematic sample of 167 primary sampling units was selected from an inventory of German communities stratified according to the BIK classification system that measures the level of urbanisation and the geographic distribution [48]. The probability of any community being picked was proportional to the number of inhabitants younger than 18 in that community. Second, an age-stratified sample of 28,400 randomly selected children and adolescents was drawn from the official registers of local residents. 17,641 participated in the KiGGS Baseline study (62.1%) between 2003 and 2006. At the second measurement point (KiGGS Wave 1 study) a total of 12,368 children and adolescents participated [50]. 6,076 out of those 12,368 participants were randomly assigned to MoMo Wave 1. The final number of participants aged 4–17 years in MoMo Wave 1 was 3,994. After exclusion of children who do not attend primary or secondary schools (mainly aged 4 and 5), a total of 1,388 primary and 2,117 secondary school children remained, building the final sample of this cross-sectional study.

### Data collection

MoMo Wave 1 data was collected between 2009 and 2012. In the MoMo Study data on physical activity was collected at central locations at the aforementioned 167 stratified sample points in Germany which were close to the participants' homes. In order to avoid systematic bias in the study results by regional or seasonal trends, the sequence of sample points visited for data collection was laid down beforehand in a random route planning. After being approached by an information letter and providing written informed consent, the children and adolescents were examined in the presence of a qualified interviewer on site [47]. Children and adolescents answered a questionnaire on their physical activity behaviors and on social environmental aspects (up to the age of 11 they did so with the help of their parents). The survey was conducted in German language.

Participation in both surveys was voluntary and anonymous and participants were informed about data security regulations prior to the investigation. Data on socio-demographics (socioeconomic status, migration background) was obtained in the KiGGS Wave 1 survey by means of telephone-based interviews. Both parents of children and adolescents up to age 17 as well as their children (from age 11) were interviewed. The survey was administered by a German-language interviewer.

### Measures

**Physical activity participation.** The MoMo Physical Activity Questionnaire (MoMo-PAQ) was used to assess self-reported habitual physical activity in different domains (physical activity in sports clubs, leisure-time physical activity outside of sports clubs, physical education, extracurricular physical activity, outdoor play, active commuting to school) in children and adolescents [51]. The MoMo-PAQ consists of 28 items and measures different domains of physical activity in a normal week, without a defined reference period. Data obtained with the MoMo-PAQ are moderately reliable (test-retest reliability: $ICC$ = 0.68) [52]. The original version of the MoMo-PAQ is available elsewhere [51, 53].

Participants were asked if they regularly participate in physical activity in sports clubs. They could list up to four different physical activities in sports clubs they regularly engage in. A dichotomous variable "physical activity in sports clubs" was built according to 1–"regular participation in physical activities in sport clubs" or 0–"no physical activities in sport clubs".

Additionally, participants were asked to if they regularly participate in physical activities outside of sports clubs (e.g., playing soccer with friends, jogging, or Inline skating). They could state up to four unorganised, leisure-time physical activities taking part outside of sports clubs. A dichotomous variable "physical activity outside of sports clubs" was built according to 1–"regular participation in physical activities outside of sports clubs" or 0–"no physical activities outside of sports clubs".

Extra-curricular physical activity participation was assessed by a question about whether the participants attend in extra-curricular physical activities. Extra-curricular physical activities are common in German schools. For example they include soccer, dancing, or ball sport courses for which interested school children can apply voluntarily. These courses usually take part every week and the attendees take part on a regular basis. For data analysis a dichotomous variable "extra-curricular physical activity" was built according to 1–"regular participation in extra-curricular physical activities" or 0–"no extra-curricular physical activities".

Unorganised outdoor play was assessed by an 8-scaled item about days per week in which the child or adolescent plays outside ("How often do you normally play outside during a week (for example: playing tag, skipping rope or going to the swimming pool)?"). A dichotomous variable "outdoor play" was built according to 1–"four or more days per week with outdoor play" or 0–"no to three days per week with outdoor play".

Commuting to school was assessed by a question about how the children and adolescents commute to school most of the time [7]. A dichotomous variable "active commuting to school" was built according to 1–"by foot or by bike, pedal-scooter or other active commuting modes" or 0–"by motor vehicle, train, bus or other inactive commuting".

**Parental and peer modelling.** Paternal and maternal modelling were measured by two single items (e.g., "Does your father regularly do sports?") which had a dichotomous answer format of 1–„yes"and 0–„no". Peer modelling was also measured by a single item („How many of your friends regularly do sports?") with a four-point rating scale ranging from 1–"none" to 4–"most of my friends".

**Parental and peer support.** Parental support scales followed the theoretical basis from Uchino [54] who postulated four functions of social support encompassing emotional, instrumental/tangible, informational, and companionship support for being related to physical activity. Each scale included the mean score (possible range: 1–4) of two items which were based on a four-point rating scale (e.g., for emotional support: "How important is sport in your family?" 1– "not at all" to 4– "very important"). Peer support included the mean score (possible range: 1–4) of a scale containing three items which were also based on a four-point rating scale (e.g., "How often do your friends ask you if you want to play outside or do sport with them (e.g., playing soccer. Riding a bicycle. Inline skating)?" 1– "never" to 4– "always"). The scales on social support had good or moderate test-retest reliability over a period of one week (test-retest reliability: $ICC_{parental\ support}$ = 0.83; $ICC_{peer\ support}$ = 0.67) [53]. The social support scales and further information on their validity and reliability are presented elsewhere [53].

**Confounding factors (sociodemographic correlates).** A migration background was assumed if the participant themselves had immigrated to Germany, if at least one parent was not born in Germany, or if both parents immigrated to Germany or had no German nationality [55]. Individual-level socioeconomic status (SES) was derived separately for both parents and included items on educational and professional status and the total household income

[56]. The higher of the two parental scores was used for analysis. Participants with separated parents were assigned the socioeconomic status of the parent they lived with. All three aspects income, educational and professional status were scored on a scale from 1 to 7 and a sum score was created (range: 3–21) and categorized into low (3–8), medium (9–14) and high (15–21) socioeconomic status [57]. The type of residential area was defined according to the number of residents living in the participant's hometown differentiated in rural area (<5,000 residents), small town (5,000–19,999 residents), medium-sized town (20,000–99,999 residents) and city (>99,999 residents). Additionally, the "region in Germany" (former East and West Germany) was captured.

## Statistical analysis

All analyses were conducted with SPSS Version 25. Socio-demographic characteristics were analysed using descriptive statistics (mean and standard deviation for continuous variables and frequency in percentage for categorical variables). Chi$^2$-tests and t-tests were used to determine gender differences in social support, social modelling and PA outcome variables.

In order to analyse the different effects of parental and peer modelling and support on variables of domain-specific physical activity participation, multiple separate logistic regressions with the different dichotomous physical activity variables as dependent variables, the modelling and support variables as correlates and age, socioeconomic status, region in Germany, residential area, and migration background as confounders were run. From these logistic regressions, odds ratios (OR) which express the influence of the different modelling and support scales on the fact whether a participant is active or not were obtained. Higher levels express a higher chance to be active in the specific domain with higher amounts of social support or positive modelling.

## Results

### Sample description

Data from 3,505 (1,788 girls, 1,717 boys) children and adolescents was eligible for the analysis in the current study. The age of the children and adolescents participating in the study ranged from 6 to 17 years, with a mean of 11.97 (SD = 3.26) years. Further information on socio-demographic characteristics of the study sample is presented in Table 1.

### Social support and social modelling in relation to gender

Gender differences in social support and social modelling are presented in Table 2. In all categories the mean value of social support was higher in primary school children than in secondary school children. In relation to gender, boys received more peer support than girls in primary as well as secondary school children. Furthermore, boys in primary school had higher levels of parental companionship support and in secondary school higher levels of parental emotional and informational support than girls. Additionally, peer modelling was higher in boys than in girls in both school levels. No gender differences were found in parental modelling. A higher percentage of maternal modelling was found than paternal modelling.

Gender differences were also found in physical activity participation in sports clubs in both primary and secondary school children, with girls being less often physically active in sports clubs than boys. Furthermore, in secondary school children, girls were less likely to be physically active outside of sports clubs, in extra-curricular physical activity, and in outdoor play, but not in active commuting to school.

**Table 1. Description of the study sample [n (%)].**

|  | Overall (N = 3505) | Girls (n = 1788) | Boys (n = 1717) |
|---|---|---|---|
| **School-type** | | | |
| primary school children | 1388 (39.6) | 699 (39.1) | 689 (41.1) |
| secondary school children | 2117 (60.4) | 1089 (60.9) | 1028 (59.9) |
| **Socioeconomic status** | | | |
| low | 255 (7.7) | 124 (7.3) | 131 (8.2) |
| medium | 2159 (65.4) | 1120 (66.2) | 1039 (64.7) |
| high | 885 (26.8) | 448 (26.5) | 437 (27.2) |
| **Migration background** | | | |
| yes | 216 (6.2) | 103 (5.8) | 113 (6.6) |
| no | 3289 (93.8) | 1685 (94.2) | 1604 (93.4) |
| **Residential area** | | | |
| rural area | 839 (23.9) | 413 (23.1) | 426 (24.8) |
| small town | 1122 (32.0) | 585 (32.7) | 537 (31.3) |
| medium-sized town | 979 (27.9) | 490 (27.4) | 489 (28.5) |
| city | 565 (16.1) | 300 (16.8) | 265 (15.4) |
| **Region in Germany** | | | |
| former east | 1141 (32.6) | 587 (32.8) | 554 (32.3) |
| former west | 2364 (67.4) | 1201 (67.2) | 1163 (67.7) |

Note. N = total sample size; n = group sample size

## Logistic regression analyses on the relationships of the social environment on physical activity

The results of the logistic regression analyses are presented separately in one table for each domain (Tables 3 to 7). With one exception in the models of active commuting to school, all significant relationships found in our analyses were positive relationships (the more/higher the support or modelling, the higher the chance to be physically active).

Regarding physical activity participation in sports clubs, all social support and social modelling indicators were significantly related with the outcome in both age groups in boys and in girls attending secondary schools (Table 3). In girls from primary schools, only peer support was not related to physical activity in sports clubs. The explained variance was highest in the regression model for physical activity in sports clubs in comparison to the other physical activity domains in all age and gender groups (0.45 to 0.60).

Extra-curricular physical activity participation was associated with parental support in boys and girls from primary schools and with peer support, peer modelling, and parental support in children from secondary schools (Table 4). Additionally, paternal modelling was related to extra-curricular physical activity in girls from secondary schools.

Explained variances in the models of physical activity outside of sports clubs were low (0.04 to 0.08) (Table 5). In these models, peer support was only related to extra-curricular physical activity participation in secondary school children and peer modelling with extra-curricular physical activity of boys also from secondary schools. Parental emotional support was related to physical activity participation in boys in both age groups and parental companionship support with physical activity participation in girls and boys from primary schools and girls from secondary schools. Furthermore, paternal and maternal modelling were associated with physical activity participation in secondary school girls and maternal modelling also with boys from primary schools.

**Table 2. Social support, social modelling and physical activity in different domains by gender and grade level.**

| | | Primary school-children | | | | Secondary school-children | | | | |
| | | Descriptive statistics | | | Bivariate analyses | Descriptive statistics | | | Bivariate analyses | |
| | | overall | boys | girls | t or chi$^2$ | p-value | overall | boys | girls | t or chi$^2$ | p-value |
|---|---|---|---|---|---|---|---|---|---|---|---|
| **Peer support** | Mean ± SD | 2.78 (0.55) | 2.82 (0.56) | 2.74 (0.53) | 2.55 | 0.011* | 2.59 (0.61) | 2.70 (0.59) | 2.48 (0.60) | 8.44 | <0.001*** |
| **Parental emotional support** | Mean ± SD | 3.07 (0.55) | 3.10 (0.56) | 3.05 (0.55) | 1.63 | 0.104 | 2.95 (0.65) | 3.00 (0.63) | 2.98 (0.66) | 3.84 | <0.001*** |
| **Parental informational support** | Mean ± SD | 3.10 (0.53) | 3.12 (0.54) | 3.07 (0.52) | 1.60 | 0.109 | 2.82 (0.64) | 2.86 (0.64) | 2.79 (0.64) | 2.68 | 0.007** |
| **Parental instrumental support** | Mean ± SD | 3.11 (0.76) | 3.12 (0.77) | 3.09 (0.75) | 0.78 | 0.434 | 2.86 (0.79) | 2.89 (0.79) | 2.83 (0.79) | 1.66 | 0.098 |
| **Parental companionship support** | Mean ± SD | 2.53 (0.56) | 2.58 (0.58) | 2.48 (0.54) | 3.37 | <0.001*** | 2.06 (0.68) | 2.09 (0.70) | 2.03 (0.66) | 1.86 | 0.064 |
| **Peer modelling** | Mean ± SD | 3.07 (0.78) | 3.12 (0.78) | 3.03 (0.78) | 2.04 | 0.042* | 3.16 (0.79) | 3.32 (0.73) | 3.02 (0.82) | 8.81 | <0.001*** |
| **Paternal modelling** | % yes | 47.2 | 46.8 | 47.7 | 0.09 | 0.762 | 47.9 | 48.1 | 47.8 | 0.02 | 0.886 |
| **Maternal modelling** | % yes | 52.9 | 51.7 | 54.2 | 0.87 | 0.350 | 53.5 | 51.7 | 55.2 | 2.68 | 0.102 |
| **Physical activity in sports clubs** | % participating | 66.8 | 70.7 | 62.9 | 9.61 | 0.002** | 65.3 | 71.9 | 59.0 | 38.41 | <0.001*** |
| **Extra-curricular physical activity** | % participating | 26.1 | 27.6 | 24.7 | 1.51 | 0.219 | 16.6 | 18.4 | 14.8 | 4.54 | 0.033* |
| **Physical activity outside of sports clubs** | % participating | 44.5 | 44.3 | 44.7 | 0.02 | 0.884 | 49.7 | 52.6 | 46.9 | 6.62 | 0.010* |
| **Outdoor play** | % with ±4 day/ week | 95.9 | 95.6 | 96.3 | 0.36 | 0.548 | 68.0 | 73.2 | 63.0 | 25.14 | <0.001*** |
| **Active commuting to school** | % active commuters | 56.8 | 55.5 | 58.0 | 0.89 | 0.347 | 45.0 | 44.4 | 45.6 | 0.28 | 0.599 |

Note.

*p < .05

**p < .01

***p < .001.

Concerning outdoor play peer support was significantly related to the outcome in all age and gender groups and additionally, all parental support constructs and peer modelling were also significant correlates in boys and girls from secondary schools (Table 6).

Concerning active commuting to school, in the present study paternal and maternal modelling and peer support were significantly associated with the outcome in primary school children (in boys and girls) (Table 7). Additionally, parental instrumental support was negatively related to active commuting to school in boys from primary as well as from secondary schools, indicating that the boys were less likely to actively commute to school if their parents provided more instrumental support for physical activity.

## Discussion

The present study revealed gender differences in parental and peer support and modelling for physical activity participation. Additionally, we found associations of social support and social modelling on physical activity participation in a variety of different physical activity domains in a large sample of children and adolescents from Germany. By considering gender differences and by taking physical activity in different domains into account, this study provides differentiated information on the relationship of social behaviour on physical activity in youth.

**Table 3. Relationships of social support and modelling and physical activity participation in sports clubs.**

| | Primary school children | | | | | | Secondary school children | | | | | |
| | boys | | | girls | | | boys | | | girls | | |
| | | 95% CI | | | 95% CI | | | 95% CI | | | 95% CI | |
| | OR[1] | Lower bound | Upper bound | OR[1] | Lower bound | Upper bound | OR1 | Lower bound | Upper bound | OR[1] | Lower bound | Upper bound |
|---|---|---|---|---|---|---|---|---|---|---|---|---|
| Peer support | 1.55** | 1.12 | 2.15 | 1.35 | 0.98 | 1.87 | 2.97*** | 2.28 | 3.86 | 1.83*** | 1.46 | 2.29 |
| Parental emotional support | 6.16*** | 4.15 | 9.15 | 3.79*** | 2.67 | 5.38 | 5.63*** | 4.23 | 7.52 | 3.86*** | 3.02 | 4.92 |
| Parental informational support | 6.22*** | 4.18 | 9.24 | 4.80*** | 3.30 | 6.99 | 4.69*** | 3.59 | 6.12 | 3.51*** | 2.77 | 4.44 |
| Parental instrumental support | 10.67*** | 7.22 | 15.75 | 8.36*** | 5.90 | 11.85 | 6.11*** | 4.71 | 7.92 | 5.92*** | 4.63 | 7.57 |
| Parental companionship support | 3.13*** | 2.20 | 4.46 | 1.64** | 1.19 | 2.28 | 2.88*** | 2.26 | 3.67 | 2.40*** | 1.92 | 3.02 |
| Peer modelling | 2.27*** | 1.76 | 2.93 | 2.28*** | 1.78 | 2.91 | 2.80*** | 2.27 | 3.45 | 2.72*** | 2.26 | 3.26 |
| Paternal modelling | 1.89** | 1.30 | 2.76 | 1.92*** | 1.35 | 2.73 | 2.34*** | 1.74 | 3.16 | 1.57** | 1.20 | 2.04 |
| Maternal modelling | 1.59* | 1.10 | 2.29 | 1.65** | 1.17 | 2.33 | 1.85*** | 1.38 | 2.47 | 1.51* | 1.16 | 1.96 |
| $R^2$(all predictors) | 0.60 | | | 0.53 | | | 0.45 | | | 0.47 | | |

Note. All models were adjusted for age, socioeconomic status, region in Germany, residential area, and migration background

*p < .05

**p < .01

***p < .001.

Overall, the results of this study revealed that gender and physical activity domains matter with respect to social behavioural relationships with physical activity in children and adolescents.

In the present study girls were less likely to be physically active than boys in almost all domains of physical activity. These finding was expected, given the fact that previous studies also presented gender differences in overall physical activity and MVPA that became more apparent in the transition from childhood to adolescence [5, 8, 58]. Previous reviews [59, 60] revealed that there is an annual decline in physical activity during adolescence (age 10–19 years) and that the decline has increased during the early stage of adolescence in girls. Contrary, in boys the decline in physical occurs during the later stages. The authors assumed that these findings might be an effect of sexual maturation, which usually happens earlier in girls compared to boys. However, social influences are also conceivable as a reason for gender differences and for declines in physical activity in the transition from childhood to adolescence [61]. Hence, future health promotion programs should especially focus on the early stage of adolescence in girls (up to the age of 13) and the older adolescent boys (from 13 years of age on). They should particularly prevent declines in physical activity during adolescence and aim to prevent the emergence of physical activity inequalities in adolescent boys and girls.

One reason for gender inequalities in physical activity in different domains could be the differences in social support perceived from peers and parents. The present study showed in accordance with other studies of children in grades five to eight from the US [35, 62], that girls received less social support from parents and from their peers than boys. Furthermore, in the present study the differences in levels of social support in boys and girls seem to be greater in secondary school children than in primary school children. However, there are no differences in instrumental support, indicating that parents do not differentiate between sons and

**Table 4. Relationships of social support and modelling and participation in extra-curricular physical activity.**

| | Primary school children | | | | | | Secondary school children | | | | | |
|---|---|---|---|---|---|---|---|---|---|---|---|---|
| | boys | | | girls | | | boys | | | girls | | |
| | | 95% CI | | | 95% CI | | | 95% CI | | | 95% CI | |
| | OR[1] | Lower bound | Upper bound | OR[1] | Lower bound | Upper bound | OR[1] | Lower bound | Upper bound | OR[1] | Lower bound | Upper bound |
| **Peer support** | 1.23 | 0.88 | 1.72 | 1.02 | 0.72 | 1.46 | 2.35*** | 1.71 | 3.24 | 1.74** | 1.27 | 2.38 |
| **Parental emotional support** | 1.85*** | 1.31 | 2.62 | 1.46* | 1.03 | 2.07 | 1.49** | 1.13 | 1.97 | 1.67** | 1.25 | 2.24 |
| **Parental informational support** | 1.70** | 1.19 | 2.42 | 1.64* | 1.12 | 2.39 | 1.32* | 1.01 | 1.73 | 1.37* | 1.03 | 1.83 |
| **Parental instrumental support** | 1.75*** | 1.35 | 2.27 | 1.33* | 1.03 | 1.72 | 1.36** | 1.10 | 1.69 | 1.66*** | 1.30 | 2.12 |
| **Parental companionship support** | 1.22 | 0.88 | 1.70 | 1.43* | 1.01 | 2.04 | 1.25 | 0.98 | 1.60 | 1.45** | 1.11 | 1.92 |
| **Peer modelling** | 1.21 | 0.95 | 1.56 | 1.00 | 0.77 | 1.28 | 1.50** | 1.18 | 1.92 | 1.47** | 1.17 | 1.86 |
| **Paternal modelling** | 1.18 | 0.81 | 1.73 | 1.11 | 0.76 | 1.62 | 1.34 | 0.95 | 1.89 | 1.66** | 1.15 | 2.38 |
| **Maternal modelling** | 1.25 | 0.85 | 1.83 | 1.10 | 0.75 | 1.62 | 1.03 | 0.73 | 1.45 | 0.87 | 0.68 | 1.39 |
| **$R^2$(all predictors)** | 0.14 | | | 0.09 | | | 0.12 | | | 0.11 | | |

Note. All models were adjusted for age, socioeconomic status, region in Germany, residential area, and migration background

*p < .05

**p < .01

***p < .001.

**Table 5. Relationships of social support and modelling and participation in physical activity outside of sports clubs.**

| | Primary school children | | | | | | Secondary school children | | | | | |
|---|---|---|---|---|---|---|---|---|---|---|---|---|
| | boys | | | girls | | | boys | | | girls | | |
| | | 95% CI | | | 95% CI | | | 95% CI | | | 95% CI | |
| | OR[1] | Lower bound | Upper bound | OR[1] | Lower bound | Upper bound | OR[1] | Lower bound | Upper bound | OR[1] | Lower bound | Upper bound |
| **Peer support** | 1.28 | 0.95 | 1.73 | 1.24 | 0.91 | 1.68 | 1.85*** | 1.47 | 2.33 | 1.74*** | 1.40 | 2.16 |
| **Parental emotional support** | 1.41* | 1.04 | 1.90 | 1.24 | 0.92 | 1.66 | 1.24** | 1.01 | 1.52 | 1.06 | 0.87 | 1.28 |
| **Parental informational support** | 1.35 | 1.00 | 1.83 | 1.22 | 0.90 | 1.65 | 1.12 | 0.91 | 1.37 | 1.05 | 0.86 | 1.27 |
| **Parental instrumental support** | 1.05 | 0.84 | 1.30 | 1.03 | 0.83 | 1.28 | 1.08 | 0.92 | 1.27 | 1.08 | 0.92 | 1.27 |
| **Parental companionship support** | 1.84*** | 1.36 | 2.48 | 1.59** | 1.17 | 2.16 | 1.11 | 0.92 | 1.34 | 1.26* | 1.03 | 1.53 |
| **Peer modelling** | 1.13 | 0.91 | 1.41 | 1.18 | 0.95 | 1.46 | 1.44*** | 1.20 | 1.72 | 1.02 | 0.87 | 1.19 |
| **Paternal modelling** | 1.08 | 0.77 | 1.50 | 0.96 | 0.69 | 1.33 | 1.14 | 0.88 | 1.48 | 1.37* | 1.07 | 1.77 |
| **Maternal modelling** | 1.56** | 1.12 | 2.17 | 1.35 | 0.97 | 1.88 | 1.27 | 0.98 | 1.64 | 1.40** | 1.09 | 1.80 |
| **$R^2$(all predictors)** | 0.07 | | | 0.04 | | | 0.08 | | | 0.07 | | |

Note. All models were adjusted for age, socioeconomic status, region in Germany, residential area, and migration background

*p < .05

**p < .01

***p < .001.

**Table 6. Relationships of social support and modelling and regular outdoor play.**

| | Primary school children | | | | | | Secondary school children | | | | | |
| | boys | | | girls | | | boys | | | girls | | |
| | OR[1] | 95% CI | | OR[1] | 95% CI | | OR[1] | 95% CI | | OR[1] | 95% CI | |
| | | Lower bound | Upper bound | | Lower bound | Upper bound | | Lower bound | Upper bound | | Lower bound | Upper bound |
|---|---|---|---|---|---|---|---|---|---|---|---|---|
| Peer support | 4.27*** | 1.88 | 9.72 | 5.25*** | 2.17 | 12.73 | 3.65*** | 2.47 | 4.86 | 3.81*** | 2.88 | 5.03 |
| Parental emotional support | 1.98 | 0.80 | 4.92 | 1.22 | 0.51 | 2.94 | 1.63*** | 1.28 | 2.07 | 1.39** | 1.11 | 1.74 |
| Parental informational support | 2.02 | 0.87 | 4.67 | 1.83 | 0.77 | 4.35 | 1.78*** | 1.41 | 2.25 | 1.42** | 1.13 | 1.78 |
| Parental instrumental support | 1.35 | 0.72 | 2.52 | 1.78 | 0.91 | 3.45 | 1.46*** | 1.21 | 1.76 | 1.22* | 1.01 | 1.47 |
| Parental companionship support | 1.89 | 0.79 | 4.52 | 2.16 | 0.78 | 6.00 | 1.75*** | 1.38 | 2.22 | 1.49** | 1.28 | 1.88 |
| Peer modelling | 1.77 | 0.92 | 3.43 | 1.07 | 0.55 | 2.07 | 1.65*** | 1.34 | 2.04 | 1.23* | 1.03 | 1.49 |
| Paternal modelling | 1.36 | 0.51 | 3.66 | 2.44 | 0.75 | 7.94 | 1.29 | 0.95 | 1.76 | 1.13 | 0.84 | 1.52 |
| Maternal modelling | 1.21 | 0.45 | 3.27 | 1.25 | 0.44 | 3.55 | 1.18 | 0.87 | 1.61 | 1.01 | 0.75 | 1.36 |
| R²(all predictors) | 0.27 | | | 0.25 | | | 0.31 | | | 0.42 | | |

Note. All models were adjusted for age, socioeconomic status, region in Germany, residential area, and migration background

*p < .05

**p < .01

***p < .001.

**Table 7. Relationships of social support and modelling and active commuting to school.**

| | Primary school children | | | | | | Secondary school children | | | | | |
| | boys | | | girls | | | boys | | | girls | | |
| | OR[1] | 95% CI | | OR[1] | 95% CI | | OR[1] | 95% CI | | OR[1] | 95% CI | |
| | | Lower bound | Upper bound | | Lower bound | Upper bound | | Lower bound | Upper bound | | Lower bound | Upper bound |
|---|---|---|---|---|---|---|---|---|---|---|---|---|
| Peer support | 1.41* | 1.05 | 1.90 | 1.37* | 1.00 | 1.88 | 1.06 | 0.85 | 1.32 | 1.02 | 0.83 | 1.27 |
| Parental emotional support | 1.11 | 0.83 | 1.48 | 1.12 | 0.83 | 1.51 | 0.96 | 0.78 | 1.18 | 1.18 | 0.97 | 1.44 |
| Parental informational support | 1.20 | 0.95 | 1.70 | 0.93 | 0.69 | 1.28 | 0.92 | 0.75 | 1.13 | 1.15 | 0.94 | 1.40 |
| Parental instrumental support | 0.72** | 0.58 | 0.90 | 0.81 | 0.65 | 1.02 | 0.69*** | 0.58 | 0.81 | 0.94 | 0.80 | 1.11 |
| Parental companionship support | 1.05 | 0.79 | 1.40 | 0.80 | 0.59 | 1.09 | 0.89 | 0.74 | 1.08 | 1.11 | 0.91 | 1.35 |
| Peer modelling | 1.15 | 0.93 | 1.43 | 1.21 | 0.98 | 1.51 | 0.85 | 0.71 | 1.01 | 0.97 | 0.83 | 1.14 |
| Paternal modelling | 1.40* | 1.01 | 1.95 | 1.68** | 1.20 | 2.35 | 0.96 | 0.74 | 1.24 | 1.20 | 0.93 | 1.55 |
| Maternal modelling | 1.52* | 1.10 | 2.10 | 1.70** | 1.22 | 2.37 | 1.14 | 0.88 | 1.48 | 0.83 | 0.65 | 1.08 |
| R²(all predictors) | 0.11 | | | 0.13 | | | 0.11 | | | 0.11 | | |

Note. All models were adjusted for age, socioeconomic status, region in Germany, residential area, and migration background

*p < .05

**p < .01

***p < .001.

daughters when providing instrumental support, for example, by driving their child to sports facilities or by buying them sports equipment. Nevertheless, some previous studies did not suggest that girls receive less parental support for physical activity than boys [31, 63, 64]. Another possibility is, that higher levels of physical activity in boys which have been found in many studies [5, 8, 9], require higher levels of social support, which could lead to a higher readiness of parents or peers to provide support [62]. Therefore, it is possible that boys receive more parental support for physical activity than girls because they claim for more support to conduct their activities.

Contrary to previous studies showing that mothers have a higher risk of inactivity than fathers [23, 65, 66] and that women are less active than men [67], we found no significant differences in modelling in mothers and fathers. However, our indicators on parental modelling were based on the children's and adolescents' reports. As a result these are indicators of perceived modelling and do not display the objective/real behaviour of the parents. Thus, the findings of the present study could be a result of the higher presence of mothers in the households taking care of their children and having their physical activity recognised by their children. Even if fathers were more active than mothers, this might not have been recognised by their children when they were asked about regular physical activity of their fathers and mothers. Imitation of social models is a function of awareness of social modelling behaviour and only occurs if the model has been recognised and was significant for the child [20].

However, same-sex hypothesis could not be confirmed in the present study. Imitation of same-sex models was not observed in most domains. This could be due to changing role models of mothers and fathers in Germany. Hence, family policies are changing in Germany and more families break out of traditional roles and today mothers are more often employed and fathers are more often taking care of their children and the household [68]. Consequently, role behaviour in regard to physical activities, sports and play could change and probably other mechanisms of social learning than same-sex imitation could occur. Further studies on social support and modelling in relation to physical activity should take into consideration who takes care of the child and to what extent.

With respect to peer modelling and support, boys reported having more physically active friends and perceived more peer support than girls, in line with expectations given the facts that more boys are regularly physically active than girls, and that youth tend to surround themselves with same-sex peers [69]. Additionally, in accordance with increasing differences in physical activity levels between boys and girls from childhood to adolescence, gender differences in peer modelling were more prevalent in secondary school children than in primary school children. As peer modelling and support were associated with a number of domain-specific physical activity measures, interventions targeting peer groups could be effective in promoting physical activity [70, 71].

Concerning the relationship of social support and social modelling with domain-specific physical activity, physical activity in sports clubs was most consistently related with the social environmental variables. Another study with adolescent girls, also found that parental and peer support were associated with sports club membership [72]. In Germany a high proportion of children and adolescents (42.2%– 71.6% depending on age and gender group) are members of sports clubs and are regularly playing sports in a club [73]. Thus, sports clubs play a central role in offering organised and instructed physical activity opportunities, and parents and peers are important instigators for boys and girls of all school-age groups. Lower levels of physical activity participation in sports clubs in girls, especially in older girls from secondary schools, could be traced to the lower levels in some kinds of social support they receive in comparison to boys.

With respect to physical activity outside of sports clubs such as free inline skating during leisure-time, jogging, or skating on halfpipe facilities that are not organised, the relevance of parents as providers of social support seem to give way to peers when transitioning from primary to secondary school. This is in line with previous studies, indicating a shift from the relevance of parents towards a growing significance of peers when children grow older [42, 62]. However, parental companionship and modelling of physical activity are still relevant factors in relation to physical activity outside of sports clubs in secondary school girls, indicating that parents might be important facilitators for leisure-time physical activity in adolescent girls. This is also true with regard to outdoor play which was also correlated with all parental support constructs in secondary school children. Morrissey and colleagues [74] who examined family support and non-school physical activity levels in adolescents also found relationships of family support with out of school physical activity. Thus, the importance of parents as promoters of physical activity seems to maintain in adolescence with respect to unorganised physical activities.

Furthermore, in regard to outdoor play and extra-curricular physical activity, the relevance of the social environment grows from primary to secondary school, but no differences in boys and girls were observed. In primary school children outdoor play was fostered by peers. Since parents tend to offer more independent outdoor play to their child if he or she is accompanied by a friend [75], peer support could be a facilitator of outdoor play for younger children regardless of gender. Equally in secondary school children, peers remain important supporters of outdoor play and all types of parental support were relevant factors.

As found in another study [44], peer support and parental modelling fostered walking or biking to school in boys, and girls from secondary school. Unexpectedly, parental instrumental support negatively predicted active commuting to school in boys which were the only negative associations between social environmental constructs and domain-specific physical activities in our study. However, one of the two items of the instrumental support variable included the question on how often parents drive their child to sports facilities. Obviously, parents who are willing to drive their child to sports facilities are also more likely to drive their child to school instead of recommending active modes of transport to school. Nevertheless, there is no explanation why this relationship was only found in boys.

## Strengths and limitations

The strength of the current study is the examination of the relationships of social modelling and social support on physical activities in a variety of different domains in a nationwide large sample of children and adolescents. The study showed that these relationships were different regarding different physical activity domains and the mechanisms of social influences on children's and adolescents' physical activity seem to differ between physical activities in different domains. Thus, this study contributes to a better understanding on social influences on physical activity by taking domain-specific physical activity into account instead of focusing on overall physical activity or on MVPA. Furthermore, the large sample size and the inclusion of children and adolescents with a wide age range enabled an analysis of differences between primary and secondary school children. Additionally, due to the fact that the data was drawn from a nationwide study conducted in 167 communities in Germany the results of this study have a high degree of representativeness.

Nevertheless, some limitations of the study have to be mentioned. First, the data of this study is cross-sectional and does not allow for the analysis of causal relationships. Therefore, we do not know the direction of the relationships found, and it is also possible that the physical activity behaviour of the children and adolescents influenced the social environment instead of

the other way around. Second, all data was captured from self or proxy reports and is prone to bias. For example, as in younger children parental reports were used to capture social support and social modelling, it is possible that parents misjudged the data. In addition, it is possible that the children did not perceive the same level of support and modelling as gauged by their parents. Third, only unspecified social support and modelling from parents and peers has been considered, and no domain-specific support and modelling data have been captured (e.g. parental informational support for active commuting to school or peer support for physical activity in sports clubs). However, Giles-Corti and colleagues [76] recommended measures to be behaviour- and context-specific. Furthermore, it could be relevant to take the family model and the main caregiver into account. Thus, further research should differentiate between social support from main caregivers and other relatives and should consider if the child lives in a single-parent family, in a traditional family, or in alternative family units.

## Conclusion

In conclusion, the present study provided comprehensive data from Germany on social support and social modelling of peers and parents and domain-specific physical activity of school children by taking gender into account. The results emphasised that these relationships vary by gender, age and physical activity domain and clearly indicate the need for the consideration of these aspects in future research. As stated by Giles-Corti et al. [76], research on environmental correlates of physical activity should be based on behaviour-specific measures that are used to predict context-specific behaviours. To go along with this recommendation, future research should focus on domain-specific physical activity behaviours and further use domain-specific social support and social modelling variables. As differences in social support between structured and unstructured physical activity settings have been observed by our study and by Spink and colleagues [46], these aspects should be observed in future research. Moreover, the reciprocal relationships of social environmental variables and physical activity should be considered, and longitudinal studies are necessary to get insights into the direction of the associations and the underlying mechanisms.

In interventions to promote physical activity in children and adolescents relevant providers of support and modelling should be targeted [77]. Showing that peer modelling and support were related to a number of domain-specific physical activity measures, interventions including peer groups could be promising in the promotion of physical activity [70, 71]. For the participation in organized sport activities in sport clubs, parents and peers are important providers of support and modelling in all age groups and for both genders and, thus, should be targeted in intervention programs. With regards to gender, interventions encompassing the social environment are required to break through gender norms and gendered cultures that neglect girls' physical activity needs and provide insufficient support for physical activity to girls–especially in the secondary school age. Same-sex hypotheses postulating social learning by focussing on same-sex models has not been confirmed in the present study.

## Author Contributions

**Conceptualization:** Anne K. Reimers, Steffen C. E. Schmidt, Yolanda Demetriou.

**Data curation:** Steffen C. E. Schmidt.

**Formal analysis:** Anne K. Reimers, Steffen C. E. Schmidt.

**Funding acquisition:** Alexander Woll.

**Investigation:** Anne K. Reimers, Steffen C. E. Schmidt, Alexander Woll.

**Methodology:** Anne K. Reimers, Steffen C. E. Schmidt, Yolanda Demetriou, Alexander Woll.

**Project administration:** Steffen C. E. Schmidt, Alexander Woll.

**Resources:** Anne K. Reimers, Alexander Woll.

**Software:** Anne K. Reimers, Steffen C. E. Schmidt.

**Supervision:** Anne K. Reimers, Alexander Woll.

**Visualization:** Steffen C. E. Schmidt, Isabel Marzi.

**Writing – original draft:** Anne K. Reimers, Isabel Marzi.

**Writing – review & editing:** Anne K. Reimers, Steffen C. E. Schmidt, Yolanda Demetriou, Isabel Marzi, Alexander Woll.

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
