## [Decision Letter · Decision Letter 0]

5 Sep 2019

[EXSCINDED]

PONE-D-19-17332

Parental and peer support and modelling in relation to domain-specific physical activity in boys and girls from Germany

PLOS ONE

Dear Prof. Dr. Reimers,

Thank you for submitting your manuscript to PLOS ONE. After careful consideration, we feel that it has merit but does not fully meet PLOS ONE’s publication criteria as it currently stands. Therefore, we invite you to submit a revised version of the manuscript that addresses the points raised during the review process.

We ask the authors to pay particular attention to the comments on statistical analysis and to the fact that the data underlying the findings in the manuscript should be fully available.

We would appreciate receiving your revised manuscript by Oct 20 2019 11:59PM. To enhance the reproducibility of your results, we recommend that if applicable you deposit your laboratory protocols in protocols.io, where a protocol can be assigned its own identifier (DOI) such that it can be cited independently in the future. For instructions see: http://journals.plos.org/plosone/s/submission-guidelines#loc-laboratory-protocols

We look forward to receiving your revised manuscript.

Kind regards,

Anne Vuillemin

Academic Editor

PLOS ONE

Journal Requirements:

3. Please include additional information regarding the survey or questionnaire used in the study and ensure that you have provided sufficient details that others could replicate the analyses; in particular , please describe in more detail how different types of parental support were assessed . If you developed a questionnaire as part of this study and it is not under a copyright more restrictive than CC-BY, please include a copy, in both the original language and English, as Supporting Information.

Additional Editor Comments (if provided):

Reviewers' comments:

Reviewer's Responses to Questions

**Comments to the Author**

1. Is the manuscript technically sound, and do the data support the conclusions?

Reviewer #1: Yes

Reviewer #2: Yes

2. Has the statistical analysis been performed appropriately and rigorously? 

Reviewer #1: No

Reviewer #2: Yes

3. Have the authors made all data underlying the findings in their manuscript fully available?

Reviewer #1: No

Reviewer #2: Yes

4. Is the manuscript presented in an intelligible fashion and written in standard English?

Reviewer #1: Yes

Reviewer #2: Yes

5. Review Comments to the Author

Reviewer #1: Introduction

Really nice integration of theory in the background. Overall nice review of literature.

Page 3, line 43 - adiposity is not a disease, it is a risk factor

Page 4, line 66 - change "probably" to "likely" or another word

Page 4, line 67 - change "typically" to "typical"

Page 5, line 95 - change "form" to "from"

Page 5, line 96 - spell out MVPA first time

Page 6, lines 119-120 - what is meant by different domains and domain-specific? This is explained in the methods, but should be introduced in this section as well

Material and Methods

Page 7, line 147 - remove comma after children

Page 8, line 170-171 - say this in the intro as examples of domains (same as comment above)

Page 8-9 - for PA in sports clubs, PA outside sports clubs, and extra-curricular PA, why were these variables dichotomized rather than used as continuous? The cutoff of 0 vs 1 or more minutes seems very arbitrary. Is this how these variables have been used in the past? I would suggest using these variables as continuous and linear rather than logistic regression for these three outcomes.

Page 14 - "...girls had a lower prevalence in physical activity..." - prevalence is not the right word here

Page 15 - Table 2 - Revise title - "Social support, social modeling, and domain-specific physical activity by gender and grade level"

Page 15 - Table 2 - Peer support, parental support, and peer modeling - doesn't make sense to me to show these as means/SD, I think you should show percents in each category as these are categorical variables. PA in sports clubs, PA outside sports clubs, and extra-curricular PA should be shown as mean number of minutes rather than "% yes". For outdoor play, if it is going to be shown as dichotomous, specify that this is the percent who said 4+ days/week.

Tables 3-5 - should be multivariate linear regression

Discussion, Strengths/limitations, conclusion

First paragraph, first sentence - split into more than one, it is long right now

Second paragraph, first sentence - revise, it is unclear

Page 28, last sentence - should it say shift rather than swift?

Page 30, last sentence - Corti, not Gorti

Add more implications for intervention work in the discussion and/or conclusion, or even a separate section

Reviewer #2: General Comment: This is an interesting study conducted with a large number of adolescents from Germany. However some aspects can still be improved in the manuscript and there are some doubts that can be better clarified throughout the text.

Comment: In the results section in the abstract, enter more information with the continuous values of the main results.

Comment: The introduction is too long, some paragraphs could be reduced to make reading more dynamic.

Comment: In the introduction, it needs to be made clear where this study is progressing compared to articles on this topic previously published in the literature.

Comment: Please insert in the methods the sample size calculation used for this study.

“Extra-curricular physical activity was assessed by a question about whether the participants attend in extra-curricular physical activities with frequency, type of activity and duration. A dichotomous variable “extra-curricular physical activity” was built according to 1–“one or more minutes extra-curricular physical activities” or 0–“no extra-curricular physical activities”.

Comment: What types of extracurricular physical activities would be considered? Please explain this further in the methods section.

Comment: A question in the methods section, did only the teens answer the questionnaires or did the teens' parents also answer some questions about social support for their children's physical activity? If so, this needs to be better addressed in the methods section.

“Participants with separated parents were assigned the socioeconomic status of the parent they lived with. All three as pects income, educational and professional status were scored on a scale from 1 to 7 and a sum score was created (range: 3–21) and categorized into low (3–8), medium (9–14) and high (15–21) socioeconomic status [57].”

Comment: This is an important aspect in the present study, please, would like the authors to present the analysis considering children of parents living together and in the other analysis children of separated parents. My question is whether children of parents living together could be more physically active when bought from children of separated parents?

Comment: In the results section, would the authors have social support information regarding parental gender? If so, it would be pertinent to analyze this relationship stratified: example: fathers x daughters; fathers x son; mothers x son; mothers x daughters.

“These findings were expected, given the fact that previous studies also presented gender differences in overall physical activity or moderate to vigorous physical activity that became more apparent in the transition from childhood to adolescence [6,9,58,59].”

Comment: Why would this happen? The authors need to advance in this aspect in the discussion.

“Another possibility is, that higher levels of physical activity in boys which have been found in many studies [5,6,9], require higher levels of social support, which could lead to a higher readiness of parents or peers to provide support [60]. Therefore, it is possible that boys receive more parental support for physical activity than girls because they claim for more support to conduct their activities”.

Comment: Would not the practice of previous physical activity of parents in their own childhood and adolescence be a factor to be considered in this relationship? Would the more physically active father throughout his life have a greater chance of his son being more physically active? So would the relationship between mothers and daughters? There is information in this sense in the literature that could be inserted as an important aspect in the discussion.

Comment: What are the practical applications of this study?

6. PLOS authors have the option to publish the peer review history of their article (what does this mean?). If published, this will include your full peer review and any attached files.

Reviewer #1: No

Reviewer #2: No

---

## [Author Response · Author response to Decision Letter 0]

11 Sep 2019

Response to review: Ms. No. PONE-D-19-17332

PLOS ONE

Dear Colleagues,

Please find enclosed our revision of the following research article:

“Parental and peer support and modelling in relation to domain-specific physical activity participation in boys and girls from Germany”

We would like to thank the reviewers for their comments. We believe that the quality of our manuscript has improved by implementing the suggestions and comments received. We hope we have addressed all the reviewers’ concerns and comments to your satisfaction.

Please find our response to the comments below. The reviewers’ comments are shown in regular font, and our responses are shown below each comment in italic:

Reviewer #1:

Introduction:

1. Really nice integration of theory in the background. Overall nice review of literature.

Thank you.

2. Page 3, line 43 - adiposity is not a disease, it is a risk factor

We deleted adiposity from the list of diseases.

3. Page 4, line 66 - change "probably" to "likely" or another word

Done.

4. Page 4, line 67 - change "typically" to "typical"

Done.

5. Page 5, line 95 - change "form" to "from"

Done.

6. Page 5, line 96 - spell out MVPA first time

Done.

7. Page 6, lines 119-120 - what is meant by different domains and domain-specific? This is explained in the methods, but should be introduced in this section as well

We added some more information and examples.

Material and Methods

8. Page 7, line 147 - remove comma after children

Done.

9. Page 8, line 170-171 - say this in the intro as examples of domains (same as comment above)

See above.

10. Page 8-9 - for PA in sports clubs, PA outside sports clubs, and extra-curricular PA, why were these variables dichotomized rather than used as continuous? The cutoff of 0 vs 1 or more minutes seems very arbitrary. Is this how these variables have been used in the past? I would suggest using these variables as continuous and linear rather than logistic regression for these three outcomes.

We agree with you and think, it could be also interesting to investigate the relationship of social environmental factors with the amount (level) of physical activity. However, it is also possible to consider physical activity participation as a dichotomous variable (1) and we decided to focus on whether or not the children and adolescents participated in physical activity in different domains due to various reasons: we focused on a large number of different domains of physical activity of which some are commonly measured in a dichotomous way (active vs. non-active; participation vs. non-participation). For example, in active transportation research it is very common to investigate whether a person is actively commuting or not (2-4) and research on correlates/determinants of active transportation focuses on the question which factors determine whether a person is an active or passive commuter. Regarding extra-curricular physical activity there is a similar situation (5-7). However, previous research on overall sports or physical activity participation also used dichotomised PA items (8-11). In addition, on a theoretical basis this is based on the assumption that there are different correlates/determinants of sports/physical activity participation (active vs. non-active) and of the amount of daily/weekly exercise/physical activity. Our paper focusses on participation and we revised the manuscript to make this even more conspicuous throughout the manuscript. 

11. Page 14 - "...girls had a lower prevalence in physical activity..." - prevalence is not the right word here

We revised this sentence.

12. Page 15 - Table 2 - Revise title - "Social support, social modeling, and domain-specific physical activity by gender and grade level"

Done.

13. Page 15 - Table 2 - Peer support, parental support, and peer modeling - doesn't make sense to me to show these as means/SD, I think you should show percents in each category as these are categorical variables.

Our social support scales are based on more than one item (2 or 3) and thus, contain calculated mean values. That is why we presented the mean/SD in our tables as also done in other studies before (12-14).

14. PA in sports clubs, PA outside sports clubs, and extra-curricular PA should be shown as mean number of minutes rather than "% yes". For outdoor play, if it is going to be shown as dichotomous, specify that this is the percent who said 4+ days/week.

We revised the column with the information on the units given in the table. See also our answer to your comment no. 10.

15. Tables 3-5 - should be multivariate linear regression

See our answer to your comment no. 10.

Discussion, Strengths/limitations, conclusion

16. First paragraph, first sentence - split into more than one, it is long right now

Done.

17. Second paragraph, first sentence - revise, it is unclear

We revised the sentence.

18. Page 28, last sentence - should it say shift rather than swift?

We corrected this typo.

19. Page 30, last sentence - Corti, not Gorti

We corrected this typo.

20. Add more implications for intervention work in the discussion and/or conclusion, or even a separate section

We added some implications for interventions in the discussion and conclusion.

Reviewer #2: 

General Comment:

21. This is an interesting study conducted with a large number of adolescents from Germany.

Thank you.

22. However some aspects can still be improved in the manuscript and there are some doubts that can be better clarified throughout the text.

Thank for your comments. We hope that, in the revised manuscript, we have addressed all your concerns and comments to your satisfaction.

Further comments:

23. In the results section in the abstract, enter more information with the continuous values of the main results.

We added some more information on the estimates of our analyses in the abstract.

24. The introduction is too long, some paragraphs could be reduced to make reading more dynamic.

We revised and shortened the introduction. However, by having the first comment from the first reviewer in mind we retained our review of the literature.

25. In the introduction, it needs to be made clear where this study is progressing compared to articles on this topic previously published in the literature.

We revised the penultimate and last paragraph of the introduction to embed our study into the state of research.

26. Please insert in the methods the sample size calculation used for this study.

The MoMo Wave 1 Study is part of the longitudinal German Health Interview and Examination Survey for Children and Adolescents, KiGGS. The survey aims to investigate and monitor health and health behaviour in children and adolescents from Germany and encompasses over 17,000 and 12,000 children and adolescents in the baseline and wave 1 data collection, respectively. In the MoMo Study as a module study of KiGGS a large proportion of the KiGGS participants could participate (in wave 1: nearly 4,000). For our research question we analysed the data from the MoMo participants that was available. Thus, no specific sample size calculation has been conducted for the research question at hand.

27. “Extra-curricular physical activity was assessed by a question about whether the participants attend in extra-curricular physical activities with frequency, type of activity and duration. A dichotomous variable “extra-curricular physical activity” was built according to 1–“one or more minutes extra-curricular physical activities” or 0–“no extra-curricular physical activities”. What types of extracurricular physical activities would be considered? Please explain this further in the methods section.

We added some more information.

28. A question in the methods section, did only the teens answer the questionnaires or did the teens' parents also answer some questions about social support for their children's physical activity? If so, this needs to be better addressed in the methods section.

No, the teens answered the questionnaires without the parents as stated in the methods section (Data collection). Just in young children up to the age of 11 had the help of their parents (but the parents did not answer some questions without the children). For a better understanding, we improved the wording in this paragraph.

29. “Participants with separated parents were assigned the socioeconomic status of the parent they lived with. All three aspects income, educational and professional status were scored on a scale from 1 to 7 and a sum score was created (range: 3–21) and categorized into low (3–8), medium (9–14) and high (15–21) socioeconomic status [57].” This is an important aspect in the present study, please, would like the authors to present the analysis considering children of parents living together and in the other analysis children of separated parents. My question is whether children of parents living together could be more physically active when bought from children of separated parents?

We agree with you that this could be a very interesting and relevant question. However, we do not have in depth data on the family model and the main caregiver. Thus, we included this relevant as an implication in the discussion section.

30. In the results section, would the authors have social support information regarding parental gender? If so, it would be pertinent to analyze this relationship stratified: example: fathers x daughters; fathers x son; mothers x son; mothers x daughters. 

Unfortunately, we do not have data on parental support for mothers and fathers separately.

31. “These findings were expected, given the fact that previous studies also presented gender differences in overall physical activity or moderate to vigorous physical activity that became more apparent in the transition from childhood to adolescence [6,9,58,59].” Why would this happen? The authors need to advance in this aspect in the discussion.

We revised this paragraph and gave some deeper insights into this topic.

32. “Another possibility is, that higher levels of physical activity in boys which have been found in many studies [5,6,9], require higher levels of social support, which could lead to a higher readiness of parents or peers to provide support [60]. Therefore, it is possible that boys receive more parental support for physical activity than girls because they claim for more support to conduct their activities”. Would not the practice of previous physical activity of parents in their own childhood and adolescence be a factor to be considered in this relationship? Would the more physically active father throughout his life have a greater chance of his son being more physically active? So would the relationship between mothers and daughters? There is information in this sense in the literature that could be inserted as an important aspect in the discussion.

This is a very relevant and interesting point. We agree with you that fathers are more physically active than mothers (as stated in our manuscript in the following paragraph). However, in our study girls reported similar (in case of paternal modelling rated by secondary school-children) or higher levels of maternal and paternal modelling than boys (see table 2). Boys did not perceive to have more active fathers than girls did and additionally, boys did not perceive to have more active same-sex models than girls.

However, we revised the paragraph to sufficiently deal with this important point.

33. What are the practical applications of this study?

See our answer to comment no. 20 of reviewer #1.

1. Welk GJ. Physical activity assessments for health-related research. Champaign, Ill.: Human Kinetics; 2002.

2. Pabayo R, Gauvin L, Barnett TA. Longitudinal Changes in Active Transportation to School in Canadian Youth Aged 6 Through 16 Years. Pediatrics. 2011;128(2):E404-E13.

3. Titze S, Giles-Corti B, Knuiman MW, Pikora TJ, Timperio A, Bull FC, et al. Associations Between Intrapersonal and Neighborhood Environmental Characteristics and Cycling for Transport and Recreation in Adults: Baseline Results From the RESIDE Study. J Phys Act Health. 2010;7(4):423-31.

4. Bere E, van der Horst K, Oenema A, Prins R, Brug J. Socio-demographic factors as correlates of active commuting to school in Rotterdam, the Netherlands. Prev Med. 2008;47(4):412-6.

5. Ara I, Vicente-Rodriguez G, Perez-Gomez J, Jimenez-Ramirez J, Serrano-Sanchez JA, Dorado C, et al. Influence of extracurricular sport activities on body composition and physical fitness in boys: a 3-year longitudinal study. Int J Obes. 2006;30(7):1062-71.

6. Gracia-Marco L, Tomas C, Vicente-Rodriguez G, Jimenez-Pavon D, Rey-Lopez JP, Ortega FB, et al. Extra-curricular participation in sports and socio-demographic factors in Spanish adolescents: the AVENA study. J Sports Sci. 2010;28(13):1383-9.

7. La Torre G, Masala D, De Vito E, Langiano E, Capelli G, Ricciardi W, et al. Extra-curricular physical activity and socioeconomic status in Italian adolescents. BMC Public Health. 2006;6:9.

8. Mader U, Martin BW, Schutz Y, Marti B. Validity of four short physical activity questionnaires in middle-aged persons. Med Sci Sports Exerc. 2006;38(7):1255-66.

9. Mota J, Almeida M, Santos P, Ribeiro JC. Perceived Neighborhood Environments and physical activity in adolescents. Prev Med. 2005;41(5-6):834-6.

10. Manz K, Mensink GBM, Jordan S, Schienkiewitz A, Krug S, Finger JD. Predictors of physical activity among older adults in Germany: a nationwide cohort study. BMJ Open. 2018;8(5):e021940.

11. Best K, Ball K, Zarnowiecki D, Stanley R, Dollman J. In Search of Consistent Predictors of Children's Physical Activity. Int J Environ Res Public Health. 2017;14(10).

12. Mutz M, Albrecht P. Parents' Social Status and Children's Daily Physical Activity: The Role of Familial Socialization and Support. Journal of Child and Family Studies. 2017;26(11):3026-35.

13. Eime RM, Casey MM, Harvey JT, Sawyer NA, Symons CM, Payne WR. Socioecological factors potentially associated with participation in physical activity and sport: A longitudinal study of adolescent girls. J Sci Med Sport. 2015;18(6):684-90.

14. Trost SG, Sallis JF, Pate RR, Freedson PS, Taylor WC, Dowda M. Evaluating a model of parental influence on youth physical activity. Am J Prev Med. 2003;25(4):277-82.

---

## [Decision Letter · Decision Letter 1]

23 Sep 2019

PONE-D-19-17332R1

Parental and peer support and modelling in relation to domain-specific physical activity participation in boys and girls from Germany

PLOS ONE

Dear Prof. Dr. Reimers,

Thank you for submitting your manuscript to PLOS ONE. After careful consideration, we feel that it has merit but does not fully meet PLOS ONE’s publication criteria as it currently stands. Therefore, we invite you to submit a revised version of the manuscript that addresses the points raised during the review process.

We would appreciate receiving your revised manuscript by Nov 07 2019 11:59PM. To enhance the reproducibility of your results, we recommend that if applicable you deposit your laboratory protocols in protocols.io, where a protocol can be assigned its own identifier (DOI) such that it can be cited independently in the future. For instructions see: http://journals.plos.org/plosone/s/submission-guidelines#loc-laboratory-protocols

We look forward to receiving your revised manuscript.

Kind regards,

Anne Vuillemin

Academic Editor

PLOS ONE

Reviewers' comments:

Reviewer's Responses to Questions

**Comments to the Author**

1. If the authors have adequately addressed your comments raised in a previous round of review and you feel that this manuscript is now acceptable for publication, you may indicate that here to bypass the “Comments to the Author” section, enter your conflict of interest statement in the “Confidential to Editor” section, and submit your "Accept" recommendation.

Reviewer #1: (No Response)

Reviewer #2: All comments have been addressed

2. Is the manuscript technically sound, and do the data support the conclusions?

Reviewer #1: Partly

Reviewer #2: Yes

3. Has the statistical analysis been performed appropriately and rigorously? 

Reviewer #1: No

Reviewer #2: Yes

4. Have the authors made all data underlying the findings in their manuscript fully available?

Reviewer #1: Yes

Reviewer #2: No

5. Is the manuscript presented in an intelligible fashion and written in standard English?

Reviewer #1: Yes

Reviewer #2: Yes

6. Review Comments to the Author

Reviewer #1: 1. I agree and you are correct that it is possible to consider PA participation as a dichotomous variable. My concern was not with the active transportation outcome variable – the way this question was asked in your data would not allow it to be used as continuous, which is why it was not included in my comment. I appreciate your examples for extracurricular and overall PA. Garcia-Marco et al. (2010) use extracurricular PA as dichotomous because this is how the question was asked in their data; La Torre et al. (2006) use a cut-off of 3 hours per week; Mota et al. (2005) assessed PA using an index which split students into four categories (sedentary, low, moderate, vigorous); and Manz et al. (2018) used a cutoff of 1 day per week and gave a reason for this cutoff. None of these examples correspond to the choice you made to use the variable as dichotomous with a cutoff of 1 minute. As stated in my previous comment, the cutoff of 0 vs 1 or more minutes seems very arbitrary. Is 1 minute of PA sufficient to be considered “participation”? If you definitely want to use these three outcomes as dichotomous, I suggest looking at recommendations or your distribution to pick a more appropriate cutoff. For example, in the U.S., there is a recommendation that adolescents participate in 60 minutes of PA every day. Since you have PA split into different types, 60 minutes is probably too high for your cutoff, but it gives you a starting point to think about it. Another starting point is to look at the mean and/or median to create two groups that are roughly the same size. Whatever you decide, the cutoff should be explained in the text and descriptive stats for these three outcomes in Table 2 should be edited accordingly.

2. This was your response to my previous comment about presenting means/SD for Peer support, parental support, and peer modeling: “Our social support scales are based on more than one item (2 or 3) and thus, contain calculated mean values. That is why we presented the mean/SD in our tables as also done in other studies before (12-14).”

You state in the manuscript: “Peer modeling was also measured by a single item (How many of your friends regularly do sports?) with a four-point rating scale ranging from 1-“none” to 4-“most of my friends”.” This is essentially a likert scale. To say that Peer modeling had a mean of 3.07 is meaningless. You should present the proportion in each of the four categories, and this variable should be an indicator variable in your model (i.e. there should be an odds ratio for each category except the reference category).

Peer support scale – Ok, so this is a peer support “score” that could range from 3-12. How is the mean below 3 if that is possible range?

Parental support scales (4 types) – Ok, so each “score” could range from 2-8. This one makes sense.

Reviewer #2: No comments, the authors answered all my questions. The quality of the work improved after the reviews performed by the authors.

7. PLOS authors have the option to publish the peer review history of their article (what does this mean?). If published, this will include your full peer review and any attached files.

Reviewer #1: No

Reviewer #2: No

---

## [Author Response · Author response to Decision Letter 1]

26 Sep 2019

Response to review: Ms. No. PONE-D-19-17332R1

PLOS ONE

Dear Colleagues,

Please find enclosed our revision of the following research article:

“Parental and peer support and modelling in relation to domain-specific physical activity participation in boys and girls from Germany”

Again, we would like to thank the reviewers for their comments. We believe that the quality of our manuscript has improved again by implementing the suggestions and comments received. We hope we have addressed all the reviewers’ concerns and comments to your satisfaction.

Please find our response to the comments below. The reviewers’ comments are shown in regular font, and our responses are shown below each comment in italic:

Reviewer #1:

1. I agree and you are correct that it is possible to consider PA participation as a dichotomous variable. My concern was not with the active transportation outcome variable – the way this question was asked in your data would not allow it to be used as continuous, which is why it was not included in my comment. I appreciate your examples for extracurricular and overall PA. Garcia-Marco et al. (2010) use extracurricular PA as dichotomous because this is how the question was asked in their data; La Torre et al. (2006) use a cut-off of 3 hours per week; Mota et al. (2005) assessed PA using an index which split students into four categories (sedentary, low, moderate, vigorous); and Manz et al. (2018) used a cutoff of 1 day per week and gave a reason for this cutoff. None of these examples correspond to the choice you made to use the variable as dichotomous with a cutoff of 1 minute. As stated in my previous comment, the cutoff of 0 vs 1 or more minutes seems very arbitrary. Is 1 minute of PA sufficient to be considered “participation”? If you definitely want to use these three outcomes as dichotomous, I suggest looking at recommendations or your distribution to pick a more appropriate cutoff. For example, in the U.S., there is a recommendation that adolescents participate in 60 minutes of PA every day. Since you have PA split into different types, 60 minutes is probably too high for your cutoff, but it gives you a starting point to think about it. Another starting point is to look at the mean and/or median to create two groups that are roughly the same size. Whatever you decide, the cutoff should be explained in the text and descriptive stats for these three outcomes in Table 2 should be edited accordingly.

We totally agree with you that 1 minute of physical activity in a specific domain does not significantly contribute to physical activity participation or health outcomes of physical activity. The cut-off of 1 or more minutes we decided is a theoretical cut-off used for our intern purposes. The physical activity items target physical activity participation. Participants were assigned to the participation categories if they indicated at least one type of physical activity they regularly conducted in the physical activity domain at hand. However, we totally agree with you that our description of the measures in the methods section was confusing and misleading. Thus, we revised this section for clarification.

Furthermore, we deeply discussed this issue in our author group and thought of possible alternative cut-offs. We were not entirely convinced to set the cut-off at the level of PA recommendations (60min) because we focus on different domains of physical activity and the recommendations postulate a minimum of overall physical activity which can be cumulated over different activities in different domains (as you already recognised). Furthermore, the mean or median is not appropriate in extra-curricular physical activity as the prevalence is too low and this cut-off would not lead to a better distribution across participation and non-participation groups. In relation to physical activity in and outside of sports clubs the cut-off nearly presents the median. From our point of view the mean or median cut-off could be less selective than the cut-off at the >0-point. And as stated above, we wanted to focus on physical activity participation and not on levels of physical activity. We made this distinction and decision, because from a theoretical and empirical perspective there is a difference between different stages of physical activity participation and between predictors of physical activity participation (active vs. non-active) and predictors of levels of physical activity (e.g., Jekauc et al., 2013; Prochaska et al., 1992; Welk, 2002). For example, different socio-demographic predictors have been found for participation in sports clubs and for levels of physical activity in sports clubs, respectively (Jekauc et al., 2013). Thus, it is necessary to decide to focus on physical activity participation or on the level of physical activity and identify the corresponding social environmental predictors.

Due to these arguments that we discussed in our author group we prefer to leave the cut-offs as set. 

2. This was your response to my previous comment about presenting means/SD for Peer support, parental support, and peer modeling: “Our social support scales are based on more than one item (2 or 3) and thus, contain calculated mean values. That is why we presented the mean/SD in our tables as also done in other studies before (12-14).”

You state in the manuscript: “Peer modeling was also measured by a single item (How many of your friends regularly do sports?) with a four-point rating scale ranging from 1-“none” to 4-“most of my friends”.” This is essentially a likert scale. To say that Peer modeling had a mean of 3.07 is meaningless. You should present the proportion in each of the four categories, and this variable should be an indicator variable in your model (i.e. there should be an odds ratio for each category except the reference category).

We absolutely understand your point here and this refers to a long consisting discussion. A Likert scale is formal categorical and should be presented by median and modus but there is also some consent in literature, that when the item is symmetrical it can be treated as equidistant and parametric tests can be used. To our knowledge, this is one important reason why symmetrical Likert scales are used that often. A summary of the discussion and some theoretical experiments can be found for example at Lantz (2013). If we argue that these items are not equidistant, we also have to argue that we cannot build sums or means of them like for the other social support scales. Thus, this issue does not only affect the peer modelling item but also the other scales. We agree with you, that this item can just barely be called symmetrical because “none” and “most of my friends” do not sound like logical ends of a spectrum at first glance. However, this wording leads to lesser skewness and kurtosis of the item, because a theoretical pole “every friend” is very hard to affirm by the participants. Thus, the struggle for equidistance and symmetry has its own story in this item. Another reason why we prefer to stick with the metrical assumption is that when we calculate the regressions with this item as categorical, there will be more than one estimate of its influence and we cannot compare the ORs between the constructs peer and parental support without further limitations. This strategy has also been adopted in other studies on social environmental influences on physical activity (Deforche et al., 2010; Shen et al, 2018; Wiium & Safvenbom, 2019).

Due to these points and with the aforementioned arguments at side, we prefer to stuck to our original approach in the current manuscript. However, we see that both options have important advantages and disadvantages (we weight the loss of comparability and consistent threat of all those Likert items high). Thus, we are willing to accept your final decision based on your expertise.

3. Peer support scale – Ok, so this is a peer support “score” that could range from 3-12. How is the mean below 3 if that is possible range?

For social support we computed the mean score of the items included in the scales. For clarification we added this information in the methods section (“Parental and peer support”).

4. Parental support scales (4 types) – Ok, so each “score” could range from 2-8. This one makes sense.

Ok, thanks.

Reviewer #2:

No comments, the authors answered all my questions. The quality of the work improved after the reviews performed by the authors.

Thank you.

Literature:

Deforche, B., Van Dyck, D., Verloigne, M., & De Bourdeaudhuij, I. (2010). Perceived Social and Physical Environmental Correlates of Physical Activity in Older Adolescents and the Moderating Effect of Self-Efficacy. Preventive Medicine, 50, S24-S29. doi:DOI 10.1016/j.ypmed.2009.08.017

Jekauc, D., Reimers, A. K., Wagner, M. O., & Woll, A. (2013). Physical Activity in Sports Clubs of Children and Adolescents in Germany: Results from a Nationwide Representative Survey. Journal of Public Health, 21(6), 505-513. doi:10.1007/s10389-013-0579-2

Lantz, B. (2013). Equidistance of Likert-type scales and validation of inferential methods using experiments and simulations. The Electronic Journal of Business Research Methods, 11(1), 16-28.

Prochaska, J. O., DiClemente, C. C., & Norcross, J. C. (1992). In Search of How People Change. Applications to Addictive Behaviors. American Psychologist, 47(9), 1102-1114. doi:10.1037//0003-066x.47.9.1102

Shen, B., Centeio, E., Garn, A., Martin, J., Kulik, N., Somers, C., & McCaughtry, N. (2018). Parental Social Support, Perceived Competence and Enjoyment in School Physical Activity. Journal of Sport and Health Science, 7(3), 346-352. doi:10.1016/j.jshs.2016.01.003

Welk, G. J. (2002). Physical Activity Assessments for Health-Related Research. Champaign, Ill.: Human Kinetics.

Wiium, N., & Safvenbom, R. (2019). Participation in Organized Sports and Self-Organized Physical Activity: Associations with Developmental Factors. International Journal of Environmental Research and Public Health, 16(4), 16. doi:10.3390/ijerph16040585

---

## [Editor Report · Decision Letter 2]

2 Oct 2019

Parental and peer support and modelling in relation to domain-specific physical activity participation in boys and girls from Germany

PONE-D-19-17332R2

Dear Dr. Reimers,

We are pleased to inform you that your manuscript has been judged scientifically suitable for publication and will be formally accepted for publication once it complies with all outstanding technical requirements.

With kind regards,

Anne Vuillemin

Academic Editor

PLOS ONE
---

## [Editor Report · Acceptance letter]

4 Oct 2019

PONE-D-19-17332R2 

Parental and peer support and modelling in relation to domain-specific physical activity participation in boys and girls from Germany 

Dear Dr. Reimers:

I am pleased to inform you that your manuscript has been deemed suitable for publication in PLOS ONE. Congratulations! Your manuscript is now with our production department. 

With kind regards,

on behalf of

Dr. Anne Vuillemin 

Academic Editor

PLOS ONE